# Escalating a Biological Dose of Radiation in the Target Volume Applying Stereotactic Radiosurgery in Patients with Head and Neck Region Tumours

**DOI:** 10.3390/biomedicines10071484

**Published:** 2022-06-23

**Authors:** Paweł Polanowski, Krzysztof Składowski, Dorota Księżniak-Baran, Aleksandra Grządziel, Natalia Amrogowicz, Jolanta Mrochem-Kwarciak, Agnieszka Pietruszka, Marek Kentnowski, Katarzyna Polanowska

**Affiliations:** 11st Radiation and Clinical Oncology Department, Maria Sklodowska-Curie National Research Institute of Oncology, Gliwice Branch, Wybrzeże Armii Krajowej 15, 44-101 Gliwice, Poland; pawel.polanowski@io.gliwice.pl (P.P.); krzysztof.skladowski@io.gliwice.pl (K.S.); dorota.ksiezniak-baran@io.gliwice.pl (D.K.-B.); agnieszka.pietruszka@io.gliwice.pl (A.P.); marek.kentnowski@io.gliwice.pl (M.K.); 2Radiotherapy Planning Department, Maria Sklodowska-Curie National Research Institute of Oncology, Gliwice Branch, Wybrzeże Armii Krajowej 15, 44-101 Gliwice, Poland; aleksandra.grzadziel@io.gliwice.pl; 3Analytics and Clinical Biochemistry Department, Maria Sklodowska-Curie National Research Institute of Oncology, Gliwice Branch, Wybrzeże Armii Krajowej 15, 44-101 Gliwice, Poland; jolanta.mrochem-kwarciak@io.gliwice.pl; 4Ophthalmology Department, St. Barbara Provincial Hospital No. 5, Plac Medyków 1, 41-200 Sosnowiec, Poland; polanowskakatarzyna@gmail.com

**Keywords:** stereotactic radiotherapy, radiosurgery boost, head and neck cancer

## Abstract

**Background:** The treatment of head and neck tumours is a complicated process usually involving surgery, radiation therapy, and systemic treatment. Despite the multidisciplinary approach, treatment outcomes are still unsatisfactory, especially considering malignant tumours such as squamous cell carcinoma or sarcoma, where the frequency of recurrence has reached 50% of cases. The implementation of modern and precise methods of radiotherapy, such as a radiosurgery boost, may allow for the escalation of the biologically effective dose in the gross tumour volume and improve the results of treatment. **Methods:** The administration of a stereotactic radiotherapy boost can be done in two ways: an upfront boost followed by conventional radio(chemo)therapy or a direct boost after conventional radio(chemo)therapy. The boost dose depends on the primary or nodal tumour volume and localization regarding the organs at risk. It falls within the range of 10–18 Gy. **Discussion:** The collection of detailed data on the response of the disease to the radiosurgery boost combined with conventional radiotherapy as well as an assessment of early and late toxicities will contribute crucial information to the prospective modification of fractionated radiotherapy. In the case of beneficial findings, the stereotactic radiosurgery boost in the course of radio(chemo)therapy in patients with head and neck tumours will be able to replace traditional techniques of radiation, and radical schemes of treatment will be possible for future development.

## 1. Background

Methods for increasing the effectiveness of treatment in patients with head and neck (H&N) tumours have been researched for fifty years. In most cases, the clinical studies conducted in this field have not achieved significant progress or breakthroughs [1,2,3,4,5,6,7,8,9,10,11,12,13]. This corresponds to trials directed on an enhancement of a tumor’s radiosensitivity and normal tissues’ radioresistance via the impact on the radiobiological phenomena taking place during a multi fraction, several-week radiotherapy [14,15,16,17,18]. Only experiences with concurrent radiochemotherapy published in the last 20 years of the previous century showed an elongation of survival, which was proven in a few meta-analyses [19,20,21].

The honing of radiotherapy techniques such as the implementation of intensity-modulated radiation therapy (IMRT), enabling the precise delivery of a radiation dose to a tumour with reducing the dose to surrounding tissues, is a technological input into the development of treatment [22,23,24,25]. Stereotactic body radiotherapy (SBRT) and stereotactic radiosurgery (SR) brought even better possibilities for safe treatment, facilitating the accumulation of very high doses in the target volume [26,27,28,29,30] and the escalation of a biologically effective dose (BED). However, an awareness of these possibilities should not overshadow the limitations; one of the major contraindications to SR is the size of the tumour due to the volume of the adjoining healthy tissues covered by a high dose, leading to serious adverse effects [31]. It appears that a large group of patients with H&N cancers are suitable for this treatment; both primary and nodal tumours are indicated as volumes suitable for dose escalation (boost to gross tumour volume—GTV_boost_). In the past, investigators have raised the subject of boosts, incorporating these methods into radical schemes of treatment, realized by external beam radiotherapy (EBRT) [32,33,34] and brachytherapy (BT) [35,36]. In the abovementioned studies, the boost was given after conventional treatment, either in one fraction (7–15 Gy) or in several fractions (18 Gy/3 fr, 16.5 Gy/3 fr). In these examples, the 3-year local control (LC) was over 70%. Al-Mamgani demonstrated the comparable efficiency of SBRT and BT boosts, whereby the SBRT boost was acknowledged as the optimal method with fewer periprocedural complications than BT. Yau demonstrated better local control outcomes after the SBRT boost than the BT boost. The tolerance of treatment with the boost in most cases was good. An unacceptable toxicity in the form of necrosis or haemorrhage was observed with the inappropriate qualification of patients with very advanced tumours with a close adjacency to organs at risk (blood vessels and brain). Therefore, to provide safe radiotherapy treatment in the H&N region, the tumour volume and its localization in relation to organs at risk should be taken into consideration instead of solely the stage of the disease. In addition to the mentioned experiences, in vitro experiments have concentrated on researching the relationship between radiobiological parameters and clinical implications. Based on the survival measurements of two rapidly growing human H&N cancer cell lines, Qi demonstrated an advantage in BED through shortening the overall treatment time (OTT) from 7 to 6 weeks. Moreover, he stated a rapid repair rate (~17 min) and fast proliferation rate (~4 days) for HNC cells and inferred that new technical possibilities and schemes of fractionation may improve tumour control [37].

Considering technological advancement, clinical attempts, and in vitro evidence, facing the challenge and implementing radiosurgery boost in radical treatment now seems feasible. This innovative project of the current study is intended to investigate the role of SR boost in combination with conventional radiotherapy in terms of the procedure’s effectiveness and its early and late side effects, which is the next step in radiotherapy’s evolution.

## 2. Methods/Design

### 2.1. Aims

Evaluation of the efficacy of the stereotactic boost applied in patients with H&N tumours.Evaluation of the safety of the stereotactic boost applied in patients with H&N tumours.

Evaluation of the influence of the stereotactic radiotherapy boost on blood parameters reflecting tumour response (interleukin 6 (Il-6), thymidine kinase (TK), Fms-related tyrosine kinase 1 (sFlt-1)), and normal tissue response (C-reactive protein (CRP)).

### 2.2. Setting of the Study

Main assumption: OTT is shorter or equal compared to standard mult-ifraction conventional radiotherapy realized to a total dose (TD) of 70 Gy in 35 fractions over 49 days.A stereotactic radiosurgery boost is given in two cases:
a.Upfront boost—on the first day of OTT (to 6 days before beginning conventional radiotherapy).
The dose of the upfront boost is prescribed on the output volume of the primary (GTV_p_) or nodal (GTV_n_) tumour at the early stages of the disease. After an upfront boost, the patient starts conventional radiotherapy to a total dose of 60 Gy.b.Direct boost—on Days 43–49 of OTT in relation to conventional radiotherapy to TD 70 Gy, i.e., up to 6 days after a dose of 60 Gy.

The dose of the direct boost is prescribed on the residual volume of GTV_p_ or GTV_n_ based on imaging technologies: positron emission tomography-computed tomography (18F-FDG PET-CT) and/or magnetic resonance imaging (MRI) performed after the audit conducted in the 5th week of conventional treatment at the advanced stages of the disease. Patients who are qualified for radical conventional radiotherapy or radiochemotherapy are prepared for conventional treatment at a total dose of 70 Gy. If they have a favorable answer after 50 Gy according to the abovementioned imaging and clinical exams, they can qualify for a direct boost. In rare cases of complete regression found based on imaging, patients finish the prescribed treatment to 70 Gy (no possibility to identify tumour volume).

3.Treatment is conducted as radiotherapy alone or radiochemotherapy based on cisplatin 100 mg/m^2^ or 40 mg/m^2^ in a 21-day or 7-day cycle, respectively. To guarantee patient safety, chemotherapy must not be taken 96 h after the stereotactic boost. Indications for radiochemotherapy are as follows:
a.Stage II-III of H&N cancer in definitive treatment.b.An extracapsular extension in dissected lymph nodes or a positive margin in a histopathological report after surgery.
4.Patients who start the treatment from induction chemotherapy (three cycles with a 21-day interval) with 75 mg/m^2^ docetaxel on Day 1, 75 mg/m^2^ cisplatin on Day 1 and 750 mg/m^2^ 5-fluorouracil by 24-h continuous infusion for 5 days (TPF), or 80–100 mg/m^2^ cisplatin on Day 1 and 800–1000 mg/m^2^ 5-fluorouracil by 24-h continuous infusion for 5 days (PF) due to a large mass of primary or nodal tumours may have a chance to qualify for an upfront or direct boost, depending the evaluation of the response to systemic treatment [38,39].5.Prescribed doses of stereotactic boost are in the range of 10–18 Gy considering the histopathological report, diameters and volumes of target, localization of the tumour in the H&N region, and adjacency of organs at risk (tolerance doses are shown in Table 1) [40].

### 2.3. Characteristics of the Participants

A group of 80 patients qualified for radical treatment in November 2019 at the Maria Sklodowska–Curie National Research Institute of Oncology in Gliwice.

### 2.4. Inclusion Criteria

Patients with squamous cell carcinoma (SCC) or adenoid cystic carcinoma (ACC) of the H&N region qualified for radical treatment with (definitive or adjuvant (adjuvant radiotherapy or radiochemotherapy in postoperative cases of R2 resection or early locoregional recurrence unsuitable for reoperation)) radiotherapy or radiochemotherapy.Patients with other malignant tumours of the H&N region (sarcomas, neuroendocrine carcinomas, differentiated carcinomas, undifferentiated carcinomas, or basaloid carcinomas) qualified for radical treatment with (definitive or adjuvant (adjuvant radiotherapy or radiochemotherapy in postoperative cases of R2 resection or early locoregional recurrence unsuitable for reoperation)) radiotherapy or radiochemotherapy.Patients with nonmalignant tumours of the H&N region (tumour mixtus or paraganglioma) demanding definitive or adjuvant radiotherapy.Age: 18–80 years.Performance status: Eastern Cooperative Oncology Group (ECOG) performance status score of 0–2.Conscious agreement to participate in the clinical trial.

### 2.5. Exclusion Criteria

Do not meet the inclusion criteria.Decompensated diabetes mellitus.Myocardial infarction occurred up to 6 months before.Pregnancy.Mental disorder preclusive of making a conscious agreement.

### 2.6. Preparations for Treatment and Planning Process

Upfront boost followed by conventional radiotherapy

Five-point head, neck, and shoulder masks are used for immobilization; computed tomography (CT) scans (SOMATOM Definition AS or SOMATOMgo.Open Pro, Siemens, Erlangen, Germany) without contrast and 1.5T MRI scans (Magnetom Aera, Siemens, Erlangen, Germany) with gadolinium intravenous contrast are performed; if there are any doubts, 18F-FDG PET-CT scans (Biograph mCT Flow40-4R or Biograph mCT X-4R, Siemens, Erlangen, Germany) are carried out in the process of preparing for an upfront stereotactic boost and conventional radiotherapy. All imaging modalities, using a spacing with 1–2 mm slice thickness, are performed in the supine position.

2.Conventional radiotherapy followed by direct boost

Five-point head, neck, and shoulder masks for patient immobilization and CT scans (3 mm slice thickness) with or without intravenous contrast (according to physician decisions) in the supine position are performed during the planning of the conventional radiotherapy process. After the audit is conducted in the 5^th^ week of conventional treatment and the qualification for a direct boost is met, a new 5-point mask, CT scan without contrast, and MRI scan with gadolinium intravenous contrast (and if there are any doubts, an18F-FDG PET-CT scan) (all with 1–2 mm slice thickness) in the supine position are performed in the planning process of the direct stereotactic boost.

The physician is responsible for contouring the target volumes (GTV; clinical target volume—CTV; planning target volume—PTV) and organs at risk (chiasm, brainstem, eyes, lenses, spinal cord, mandible, cochleae, brain, parotid glands, larynx, and thyroid gland). The medical physicist prepares conventional and stereotactic radiotherapy plans according to the International Commission on Radiation Units and Measurements (ICRU) Report No.83 guidelines [41], and the physician accepts dose distribution in target volumes and organs at risk relying on the data in the attachment (Table 1 and Table 2). Next, the accepted plan is verified in an ionization chamber in slab phantom measurements for CyberKnife planning or in electronic portal imaging devices (EPIDs) for EDGE linear accelerator. For CyberKnife patient-specific dosimetry, a 5% point dose difference is accepted. In EPID dosimetry, gamma assessment is used according to the following criteria: 3% dose difference and 2 mm distance-to-agreement and 95% agreement among all analyzed points.

The stereotactic radiosurgery boost is executed on two types of accelerators:Cyber Knife^®^ VSI or M6 series (Accuray, Sunnyvale, CA, USA)—an acceleration voltage of 6 MV, flattening filter-free (FFF) beams, and a maximal dose rate of 1000 MU/min. The kV imaging to verify the patient’s position during a therapeutic session is performed in the range of 15–150 s. The session time usually ranges from 30 to 60 min.Edge^TM^ radiosurgery accelerator (Varian Medical Systems, Palo Alto, CA, USA)—an acceleration voltage of 6 MV, a flattening filter and FFF beams, and a maximal dose rate of 1400 MU/min. Image-guided radiotherapy (IGRT) is performed before a therapeutic session using ConeBeam CT scans. The session time using volumetric modulated arc therapy (VMAT) usually ranges from 15 to 20 min.

### 2.7. Doses of Stereotactic Radiosurgery Boost

The contouring of target volumes in the planning of the conventional radiotherapy process is based on international guidelines [42,43]. The PTV margin added in planning stereotactic radiotherapy to primary or nodal GTV_boost_ is in the range of 1–3 mm, depending on the GTV_boost_ volume (Table 3).

The abovementioned doses could be changed (decreased) after considering tumour volume (GTV_boost_) depending on localization regarding organs at risk.

Table 4 consists of the biologically effective dose (BED) and equivalent dose in 2 Gy fraction (EQD2) calculations based on an assumption of an α/β ratio for SCC in the range of 10–6.5 Gy and an α/β ratio = 3 Gy for late effects [44,45].

### 2.8. Monitoring of Treatment

During the treatment, the patients are hospitalized in the I Radiotherapy and Clinical Oncology Department in the Maria Sklodowska–Curie National Research Institute of Oncology in Gliwice. Fasting blood tests (blood morphology, aspartate aminotransferase (AST), alanine aminotransferase (ALT), total bilirubin, creatinine, urea, sodium, potassium, Il-6, TK, sFlt-1, and CRP) are performed a few times during treatment in compliance with the clinical state of the patients, with at least 3 measurements depending on the scheme of the treatment:Upfront boost—before the boost, one day after the boost, and at the end of the conventional treatment.Direct boost—before the conventional treatment, one day before the boost, and one day after the boost.

With the aim of evaluating tolerance, patients are asked to complete the QLQ-C30 and H&N35 questionnaires before, during, and at the end of treatment.

### 2.9. Follow Up

Follow-up visits are performed with imaging procedures (CT scan, MRI scan, and 18F-FDG PET-CT scan) one month after the end of treatment, every 3 months in the first year, every 4 months in the second year, and every six months in the third–fifth years.

### 2.10. Endpoints Andstatistical Analysis

The primary endpoint is the response to treatment in imaging tests and clinical examination—local control (LC) and locoregional control (LRC).The secondary endpoints are as follows:
➢Evaluation of efficacy—overall survival (OS), disease-free survival (DFS), and progression-free survival (PFS).➢Evaluation of safety—acute and late side effects according to the Common Terminology Criteria for Adverse Events (CTCAE)v4.0.➢Evaluation of tolerance—QLQ-C30 and H&N35 questionnaires.
Statistical analysis: statistical modelling, regression with random effects, Kaplan–Meier estimator, and Cox and Weibull regression models will be used.

## 3. Discussion

### 3.1. Conventional Radiotherapy

Conventional radiotherapy in the treatment of all malignant tumours, beyond ACC, is performed in two steps: 50 Gy in 25 fractions as a prophylactic dose to the uninvolved lymphatic nodes in the H&N region and 60 Gy in 30 fractions in high-risk prophylactic regions, i.e., whole group(s) with metastatic lymph node(s) and the anatomical site of the primary tumour, e.g., whole oropharynx in the case of tonsil cancer.

Single-step conventional radiotherapy to a total dose of 50 Gy in 25 fractions is prescribed in patients with adenoid cystic carcinoma and nonmalignant tumours at the anatomical site of the primary tumour, along with the nerve pathways to the base of the skull in cases of ACC.

Positive HPV status does not impact the de-escalation of conventional radiotherapy or the dose of the boost. kV imaging to verify the patient’s position is performed every day before a therapeutic session.

### 3.2. Chemotherapy

To reduce the toxicity and the risk of electrolyte disorder during induction chemotherapy (TPF or PF) and concurrent radiochemotherapy, an administration of intravenous hydration is recommended: 1000 mL of 0.9% sodium chloride before and after cisplatin infusion with 100 mL of 15% mannitol, 10 mEq potassium chloride, and 10 mL of 20% magnesium sulfate. Due to the emetogenic potential of chemotherapy, the following drugs are ordered: 300 mg netupitant + 0.5 mg palonosetron one hour before chemotherapy on Day 1 and 8 mg dexamethasone (Days 1–5).

The primary prophylaxis of febrile neutropenia during the TPF scheme is realized with a recombinant granulocyte colony-stimulating factor—administering 6 mg pegfilgrastim subcutaneously via a single-dose prefilled syringe. To prevent infection during potential neutropenia after the TPF, 500 mg ciprofloxacin per os twice a day is prescribed from Day 5 to Day 15.

### 3.3. Laboratory Test Technology

Il-6 and SFlt-1 are determined using an electrochemiluminescent immunoenzymometric assay (Roche Diagnostics, Mannheim, Germany) on the Cobas e801 system.

The serum TK levels are determined using fully automated chemiluminescence technology with magnetic microparticles on the Liason XL system (DiaSorin S.p.A. Via Crescentino, snc, 13040 Saluggia VC, Italy).

The high-sensitivity C-reactive protein levels are measured using nephelometry on the Atellica system (Siemens Healthcare).

The complete blood count results are obtained using a Sysmex XN-2000 haematology analyser.

AST, ALT, total bilirubin, creatinine, urea, sodium, and potassium are measured on an Alinity c system (Abbott Diagnostics Division, Mannheim, Germany) using biochemical methods.

## Figures and Tables

**Table 1 biomedicines-10-01484-t001:** Tolerance doses in one fraction.

Serial Tissue	Volume	Volume Max (Gy)	Max Point Dose (Gy)	Endpoint (≥Grade 3)
Optic Pathway	<0.2 cc	8 Gy	10 Gy	neuritis
Cochlea			9 Gy	hearing loss
Brainstem (not medulla)	<0.5 cc	10 Gy	15 Gy	cranial neuropathy
Spinal Cord and medulla	<0.35 cc <1.2 cc	10 Gy 8 Gy	14 Gy	myelitis
Spinal Cord Subvolume (5–6 mm above and below level treated per Ryu)	<10% of subvolume	10 Gy	14 Gy	myelitis
Cauda Equina	<5 cc	14 Gy	16 Gy	neuritis
Sacral Plexus	<5 cc	14.4 Gy	16 Gy	neuropathy
Esophagus	<5 cc	11.9 Gy	15.4 Gy	stenosis/fistula
Brachial Plexus	<3 cc	13.6 Gy	16.4 Gy	neuropathy
Heart/Pericardium	<15 cc	16 Gy	22 Gy	pericarditis
Great vessels	<10 cc	31 Gy	37 Gy	aneurysm
Trachea and Large Bronchus	<4 cc	17.4 Gy	20.2 Gy	stenosis/fistula
Bronchus- smaller airways	<0.5 cc	12.4 Gy	13.3 Gy	stenosis with atelectasis
Rib	<5 cc	28 Gy	33 Gy	Pain or fracture
Skin	<10 cc	25.5 Gy	27.5 Gy	ulceration
Stomach	<5 cc	17.4 Gy	22 Gy	ulceration/fistula
Bile duct			30 Gy	stenosis
Duodenum	<5 cc <10 cc	11.2 Gy 9 Gy	17 Gy	ulceration
Jejunum/Ileum	<30 cc	12.5 Gy	22 Gy	enteritis/obstruction
Colon	<20 cc	18 Gy	29.2 Gy	colitis/fistula
Rectum	<3.5 cc <20 cc	39 Gy 22 Gy	44.2 Gy	proctitis/fistula
Ureter			35 Gy	stenosis
Bladder wall	<15 cc	12 Gy	25 Gy	cystitis/fistula
Penile bulb	<3 cc	16 Gy		impotence
Femoral Heads	<10 cc	15 Gy		necrosis
Renal hilum/vascular trunk	15 cc	14 Gy		malignant hypertension
**Parallel Tissue**	**Critical Volume (cc)**	**Critical Volume Dose Max (Gy)**		**Endpoint (≥Grade 3)**
Lung (Right & Left)	1500 cc	7 Gy		Basic Lung Function
Lung (Right & Left)	1000 cc	7.6 Gy	V-8 Gy < 37%	Pneumonitis
Liver	700 cc	11 Gy		Basic Liver Function
Renal cortex (Right & Left)	200 cc	9.5 Gy		Basic renal function

**Table 2 biomedicines-10-01484-t002:** Tolerance doses in 30 fractions (conventional fractionation).

Serial Tissue	Contouring Instructions	Volume	Volume Max (Gy)	Max Point Dose (Gy)	Endpoint (≥Grade 3)
**Optic Pathway**	One structure both sides from posterior globe, including chiasm, to proximal optic radiations	<0.5 cc	44 Gy	52 Gy	neuritis
**Eye (retina)**	Each side separately, entire globe	Mean dose	<38 Gy	45 Gy	retinitis
**Lens**	Each side separately			10 Gy	cataract
**Eyelid—Meibomian glands (one side)**	Each side separately, upper and lower lid as one structure			32 Gy	dry eye syndrome
**Lacrimal gland (one side)**	Each side separately	<1 cc	20 Gy	36 Gy	lack of tears
**Cochlea**	Each side separately, include at least 3 CT slices	<0.5 cc	36 Gy	40 Gy	hearing loss
**Brainstem (not medulla)**	Superiorly from incisura, midbrain and pons only, one structure	<2 cc	50 Gy	60 Gy	cranial neuropathy
**Spinal Cord and medulla**	For medulla: starting at inferior pons to foramen magnum. For cord: entire bony canal including at least 10 cm superior and inferior to PTV	<5 cc	44 Gy	50 Gy	myelitis
**Salivary gland (one side)**	Each parotid gland separately	<7 cc Mean dose	20 Gy <26 Gy	32 Gy	xerostomia
**Larynx**	Starting 1 cm above first appearance of true vocal cord including entire cord, arytenoid muscles, corniculate and arytenoid cartilages, and portions of thyroid cartilage abutting these structures ending at the first appearance of the cricothyroid ligament.	<3 cc	40 Gy	46 Gy	necrosis/edema
**Temporomandibular joint**	Each side separately starting at the superior articular surface near the zygoma bone and ending at the notch at the superior part of the ramus of the mandible.	<1 cc	60 Gy	65 Gy	inflammation
**Esophagus**	Include the mucosal, submucosa, and all muscular layers out to the fatty adventitia at least 10 cm superior and inferior to PTV	<5 cc	55 Gy	60 Gy	stenosis/fistula
**Brachial Plexus**	Each side separately from the spinal nerves exiting the neuroforamina from around C5 to T2 to include only the major trunks of the brachial plexus using the subclavian and axillary vessels as a surrogate for identifying its location, extending proximally at the bifurcation of the brachiocephalic trunk into the jugular/subclavian veins (or carotid/subclavian arteries), and following along the route of the subclavian vein to the axillary vein ending after the neurovascular structures cross the second rib.	<3 cc	62 Gy	66 Gy	neuropathy
**Heart/Pericardium**	Contoured along with the pericardial sac. The superior aspect (or base) for purposes of contouring will begin at the level of the inferior aspect of the aortic arch (aorto-pulmonary window) and extend inferiorly to the apex of the heart.	<15 cc	60 Gy	60 Gy	pericarditis
**Great vessels**	The wall and lumen of the named vessel at least 10 cm superior and inferior to PTV	<10 cc	60 Gy	76 Gy	aneurysm
**Trachea and Large Bronchus**	Contour the trachea and cartilage rings starting 10 cm superior to the PTV extending inferiorly to the bronchi ending at the first bifurcation of the named lobar bronchus.	<5 cc	60 Gy	66 Gy	stenosis/fistula
**Skin**	The outer 0.5 cm of the body surface anywhere within the whole body contour.	<10 cc	70 Gy	76 Gy	ulceration
**Parallel Tissue**		**Critical Volume (cc)**	**Critical Volume Dose Max (Gy)**	**Other Constraints**	**Endpoint (≥Grade 3)**
**Lung (Right & Left) minus GTV**	Contour right and left lung as one structure including all parenchymal lung tissue but exluding the GTV and major airways (trachea and main/lobar bronchi)	1500 cc	14 Gy		Basic Lung Function
**Lung (Right & Left) minus GTV**	Contour right and left lung as one structure including all parenchymal lung tissue but exluding the GTV and major airways (trachea and main/lobar bronchi)	1000 cc	15 Gy	Mean dose < 20 Gy, V-20 Gy < 37%	Pneumonitis

**Table 3 biomedicines-10-01484-t003:** Range of boost doses depending on tumour volume.

Tumour Volume GTV_boost_	Range of Stereotactic Boost Doses
≥17 cm^3^	10–12 Gy
7–16.9 cm^3^	13–15 Gy
<7 cm^3^	16–18 Gy

**Table 4 biomedicines-10-01484-t004:** Commutation of physical dose on BED and EQD2.

Physical Dose	BED α/β = 10	EQD_2_α/β = 10	BED α/β = 6.5	EQD_2_α/β = 6.5	BED α/β = 3	EQD_2_α/β = 3
**1 × 10 Gy = 10 Gy**	20.0	16.7	25.4	19.4	43.3	26.0
**1 × 12 Gy = 12 Gy**	26.4	22.0	34.2	26.1	60.0	36.0
**1 × 15 Gy = 15 Gy**	37.5	31.2	49.6	37.9	90.0	54.0
**1 × 18 Gy = 18 Gy**	50.4	42.0	67.8	51.9	126.0	75.6

## Data Availability

The datasets used and/or analyzed during the current study are available from the corresponding author on reasonable request.

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
