# Peer review of "Escalating a Biological Dose of Radiation in the Target Volume Applying Stereotactic Radiosurgery in Patients with Head and Neck Region Tumours"

_biomedicines, 2022, doi:10.3390/biomedicines10071484_

Round 1

Reviewer 1 Report

This is an interesting study protocol about escalating the biological dose of radiation in the target volume applying stereotactic radiosurgery in patients with head and neck tumours.

The paper is well written. However, some issues remain.

There are some grammatical errors. Please improve readability of the manuscript.

The authors must specify if they will analyze Human Papillomavirus infection in head and neck carcinomas. Indeed, it may represent a confounding factor in the analyses.

From which anatomical subsites will origin the included tumors?

Statistical analyses should be stratified according to histotype.

Author Response

Thank you very much for your review.

We have made effort to ensure proper readability. Before submitting this paper to the journal, we ordered professional English services to improve the quality of our publication. The certificate is attached. However, taking into account your opinion, we may ask for English services once again, if you suggest this solution.

Regarding to HPV infection, the protocol allows to include patients with HPV + or HPV - status. The identification of HPV status in patients with oropharynx cancer is standard procedure in our Institute, both in histopathological specimens and as a liquid biopsy (DNA HPV in the serum). Better prognosis of HPV-related patients and attempts to de-escalating of dose of radiation are well known for a few years but up to now, we have not got sufficient proof to deescalate treatment due to ongoing research. After collecting data considering HPV status, we will analyze HPV + and HPV- groups, separately.  

Our first experiences concern nasopharyngeal cancer, oropharynx cancer, palatine cancer (SRS boost on primary tumors), and FPI (SRS boost on metastatic nodes). We hope that we will have the opportunity to qualify patients with other localizations.

Obviously, statistical analysis will be performed with accuracy after stratification according to histotype.

Reviewer 2 Report

Dear Authors the work is well done and for me suitable for pubblication

best regards

Author Response

Thank you very much for your positive review.